# An image quality assessment index based on image features and keypoints for X-ray CT images

**Sho Maruyama**[ID]*, **Haruyuki Watanabe, Masayuki Shimosegawa**

Department of Radiological Technology, Gunma Prefectural College of Health Sciences, Maebashi, Gunma, Japan

* maruyama@gchs.ac.jp

**Data Availability Statement:** Maruyama Sho (2024). Publishing analysis data. figshare. Figure. https://doi.org/10.6084/m9.figshare.25340383.v2.

**Funding:** The author(s) received no specific funding for this work.

## Abstract

Optimization tasks in diagnostic radiological imaging require objective quantitative metrics that correlate with the subjective perception of observers. However, although one such metric, the structural similarity index (SSIM), is popular, it has limitations across various aspects in its application to medical images. In this study, we introduce a novel image quality evaluation approach based on keypoints and their associated unique image feature values, focusing on developing a framework to address the need for robustness and interpretability that are lacking in conventional methodologies. The proposed index quantifies and visualizes the distance between feature vectors associated with keypoints, which varies depending on changes in the image quality. This metric was validated on images with varying noise levels and resolution characteristics, and its applicability and effectiveness were examined by evaluating images subjected to various affine transformations. In the verification of X-ray computed tomography imaging using a head phantom, the distances between feature descriptors for each keypoint increased as the image quality degraded, exhibiting a strong correlation with the changes in the SSIM. Notably, the proposed index outperformed conventional full-reference metrics in terms of robustness to various transformations which are without changes in the image quality. Overall, the results suggested that image analysis performed using the proposed framework could effectively visualize the corresponding feature points, potentially harnessing lost feature information owing to changes in the image quality. These findings demonstrate the feasibility of applying the novel index to analyze changes in the image quality. This method may overcome limitations inherent in conventional evaluation methodologies and contribute to medical image analysis in the broader domain.

## Introduction

In diagnostic radiological imaging, it is important to optimize the balance between the radiation dose to which patients are exposed and the specific image quality required for diagnosis and operate under conditions supported by objective data [1–6]. It is desirable to use

**Competing interests:** The authors have declared that no competing interests exist.

quantitative image quality metrics that correlate with the subjective perceptions of the doctors that observe medical images and perform diagnostic decisions [7–9].

In optimization tasks for medical imaging, adjusting radiation doses is important, but also involves considerations related to image processing [6, 10]. Recent advancements in the field of diagnostic imaging have seen various algorithms implemented for various purposes, including nonlinear processing procedures [11, 12]. These processes affect the spatial correlations within the image, leading to nonlinear changes in the texture or resolution [13]. Despite limitations in applying traditional evaluation techniques to such state-of-the-art technologies, efforts to understand their characteristics in detail are significant for users [14, 15]. In addition, variations in the image quality across facilities have been reported to affect the effectiveness and reliability of detection and diagnostic tasks [16]. The significance of multicenter analysis studies has also been highlighted, as variations need to be understood and managed to enhance the consistency and overall usefulness of diagnostic imaging [17]. Thus, there is a need for metrics that can improve conventional evaluation methods and address a wide range of concerns by combining the perspective of changes in the inherent features of images with image quality evaluation [18–20].

Physical evaluation metrics such as the peak signal-to-noise ratio (PSNR), structural similarity index (SSIM), and feature similarity index (FSIM) are used extensively in medical image analysis and other fields [21–26]. These metrics are typically employed as full-reference image quality assessment (IQA), in which a high-quality reference image is compared with a target image. However, several challenges exist in that potential misalignments, shifts, scale changes, and computational parameters affect the evaluation results [27–30] and the aforementioned metrics do not adequately address these concerns. However, as full reference-type metrics can clearly express the relative quality of an image with respect to its ideal quality, they are still valuable tools.

Image features are quantified data that represent the localized information within an image, capturing the patterns and structures in local regions [31]. These features allow us to distinguish different regions within the same image and match feature points across multiple images, making them useful for pattern recognition and image matching. In diagnostic imaging, when an observer recognizes an image, the reading process progresses while associating features as either "normal" or "abnormal" with respect to the entire image or local regions [32]. Considering this insight, it can be argued that images with easily matched features exhibit superior image quality for diagnosis. That is, the image has high diagnostic value. Therefore, the distances between the feature values of the corresponding keypoints in the reference and target images can provide potentially effective quality index in medical image analysis [33]. Such a concept would address various concerns. In addition, it has the potential to be used as a simple evaluation metric that can provide straightforward and explainable interpretations.

This study focuses on developing a framework to address the need for robustness and interpretability that are currently lacking in conventional evaluation methods. We introduce a novel IQA metric approach based on keypoints and their associated unique image feature values, especially in the context of medical imaging. The main aims of this study are to evaluate the applicability and effectiveness of the proposed metric, as well as to investigate its robustness. Specifically, we evaluated the applicability of the metric by examining its performance on target images of various qualities. Furthermore, we investigated its robustness by applying various affine transformations to target images without compromising the image quality. By analyzing the correlation between the metric and the SSIM, which is regarded as an external standard for image quality evaluation, we aimed to establish the validity and effectiveness of the proposed approach. In this paper, which provides the new methodology, we demonstrate that distances between feature descriptors of corresponding keypoints can be applied provide an objective and quantitative measure of image quality consistently.

## Materials and methods

### Overview of the proposed index

The proposed IQA index (described as *PI* in this paper) considers the distance between the feature values of the corresponding keypoints in the reference and target images as a quantitative value. The measurement procedure is outlined as follows. (1) Keypoints are extracted from both the reference and target images, as shown in Fig 1(A). (2) All keypoints between the two images are compared, and pairs of corresponding keypoints are determined using brute-force matching based on the Hamming distances of the feature values (Fig 1(B)). (3) The Hamming distances between the feature values of corresponding pairs of keypoint are calculated, and the mean value of entire image is computed as the evaluation index (Fig 1(C)).

The measurement process for *PI* can be formulated as follows:

$$PI = \frac{1}{N}\sum_{i=1}^{N}\|\overrightarrow{F}_{Ri} - \overrightarrow{F}_{Ti}\|, \tag{1}$$

where $N$ denotes the number of corresponding pairs of keypoints. $\overrightarrow{F}$ is the feature vector

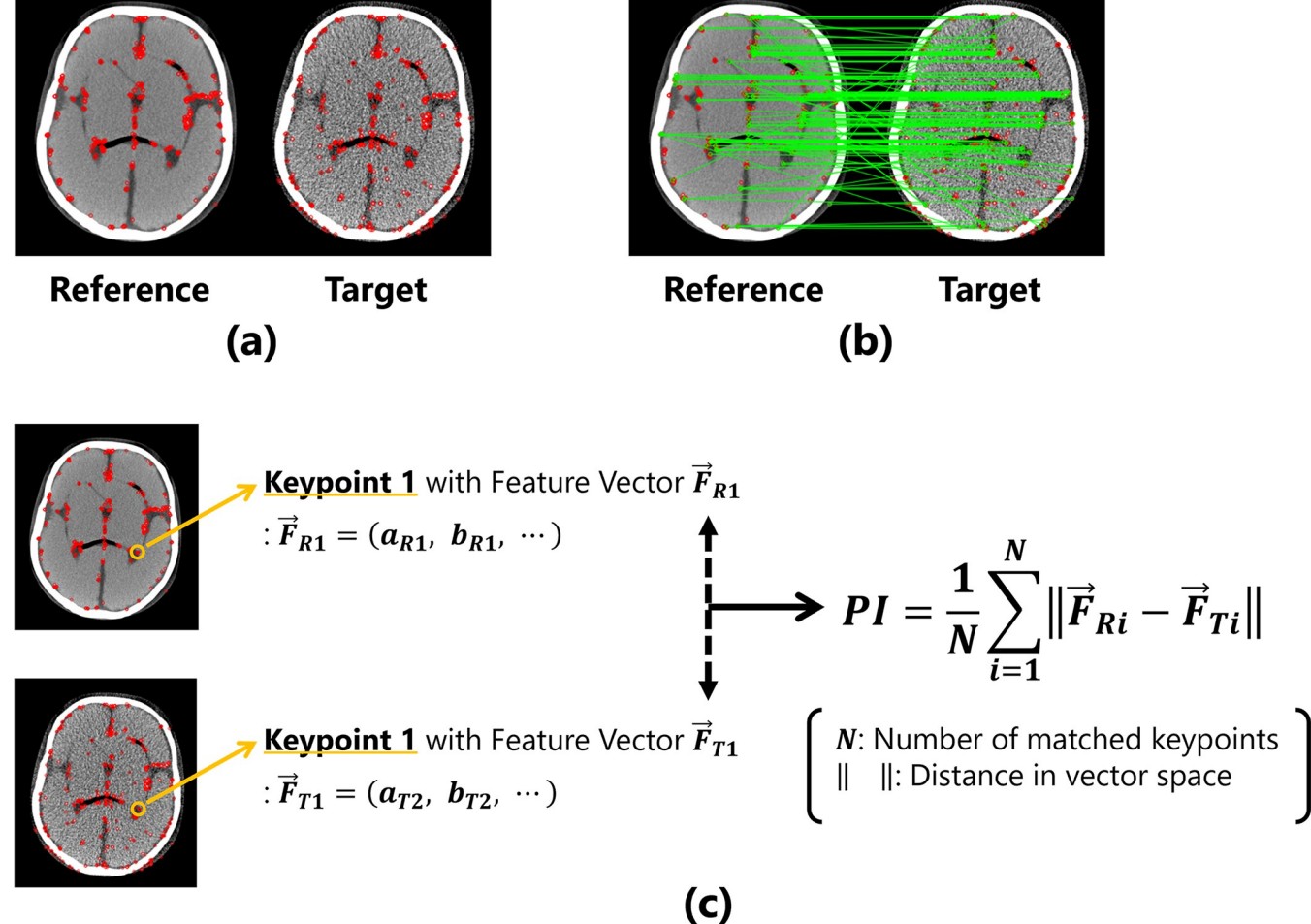

Reference    Target

**(a)**

Reference    Target

**(b)**

**Keypoint 1** with Feature Vector $\overrightarrow{F}_{R1}$

$: \overrightarrow{F}_{R1} = (a_{R1},\ b_{R1},\ \cdots)$

$$PI = \frac{1}{N}\sum_{i=1}^{N}\|\overrightarrow{F}_{Ri} - \overrightarrow{F}_{Ti}\|$$

**Keypoint 1** with Feature Vector $\overrightarrow{F}_{T1}$

$: \overrightarrow{F}_{T1} = (a_{T2},\ b_{T2},\ \cdots)$

$N$: Number of matched keypoints
$\|\ \ \|$: Distance in vector space

**(c)**

**Fig 1. Overview of the proposed method for image quality evaluation.** (a) The detected keypoints are represented by red dots. (b) Corresponding points are determined based on the features of the keypoints. (c) Calculation process under the proposed metric using keypoints and features.

possessed by each keypoint, and $\|\overrightarrow{F}_{Ri} - \overrightarrow{F}_{Ti}\|$ is the distance in the feature space. If the feature vector of the reference image is represented as $\overrightarrow{F}_R$, the feature vector of the first keypoint is:

$$\overrightarrow{F}_{R1} = (a_{R1}, b_{R1}, \cdots), \tag{2}$$

and the feature vector of the target image is also expressed as $\overrightarrow{F}_T$. For example, when the reference and target images are completely identical, the two images possess exactly the same image features, resulting in the distance between the keypoint features being 0. However, if the image quality of the target image changes and the features of the two images differ, the distance between the feature values will become large.

Image features are calculated using various algorithms such as scale invariant feature transform (SIFT) [34], speeded up robust features (SURF) [35], accelerated-KAZE (AKAZE) [36, 37], binary robust invariant scalable keypoints (BRISK) [38], and oriented features from accelerated segment test (FAST) and rotated binary robust independent elementary features (BRIEF) (ORB) [39]; FAST and BRIEF are a keypoint detection algorithm and feature description method, respectively [31]. In this verification, three types of algorithms, namely AKAZE, BRISK, and ORB, were employed for detecting the keypoints and their associated feature descriptors [40]. These feature descriptors computed binary features; the distance between the features is the Hamming distance, which is represented in Eq (1). To enhance the versatility of the proposed metrics, it would be appropriate to select more general algorithms to capture a wide range of features, rather than methodologies that are strongly suited to specific tasks. The reason for applying these general algorithms used extensively in computer vision for acquiring the proposed metrics is their high versatility and robustness [31]. With this selection, the following advantages can be expected: As they are non-machine-learning algorithms, there is no concern regarding dependency on training data; and they enable real-time processing owing to their simplicity and lack of a need for complex computations, making them effectively usable even in resource-constrained environments (implying high adaptability to hardware). In addition, despite their simplicity, they provide robust features against various image transformations, yielding stable results. The existence of such advantages is not the sole basis for the selection; these algorithms can also be implemented with the OpenCV library [40], with the aim of enhancing the accessibility of this evaluation framework.

## Data acquisition for analysis

We analyzed head images obtained using an X-ray computed tomography (CT) system. A PBU-60 head phantom (Kyoto Kagaku, Kyoto, Japan) was imaged using the Alexion CT system (Canon Medical Systems, Tochigi, Japan). The CT images were acquired under the conditions for routine head imaging in clinical practice. The tube voltage was set to 120 kV, the scan slice thickness was 0.5 mm, and the tube rotation time was 1.0 rot/s. The reconstruction field of view (FOV) was set to 250 mm with a reconstruction slice thickness and interval of 5 mm each. The reconstruction algorithm used was the filter back projection method and the reconstruction kernel employed was FC21, which is the default for head imaging. Although the imaging doses varied, as explained later, the reference image was obtained at a computed tomography dose index (CTDI-vol) of 89.1 mGy, which is higher than the diagnostic reference level in Japan (DRL2020) [41].

## Image dataset for verification

First, to verify the validity of the proposed index, we prepared a high-quality reference image obtained under the conditions described above and target images with image qualities degraded by X-ray quantum noise through reductions in the imaging dose (Fig 2(A)). The

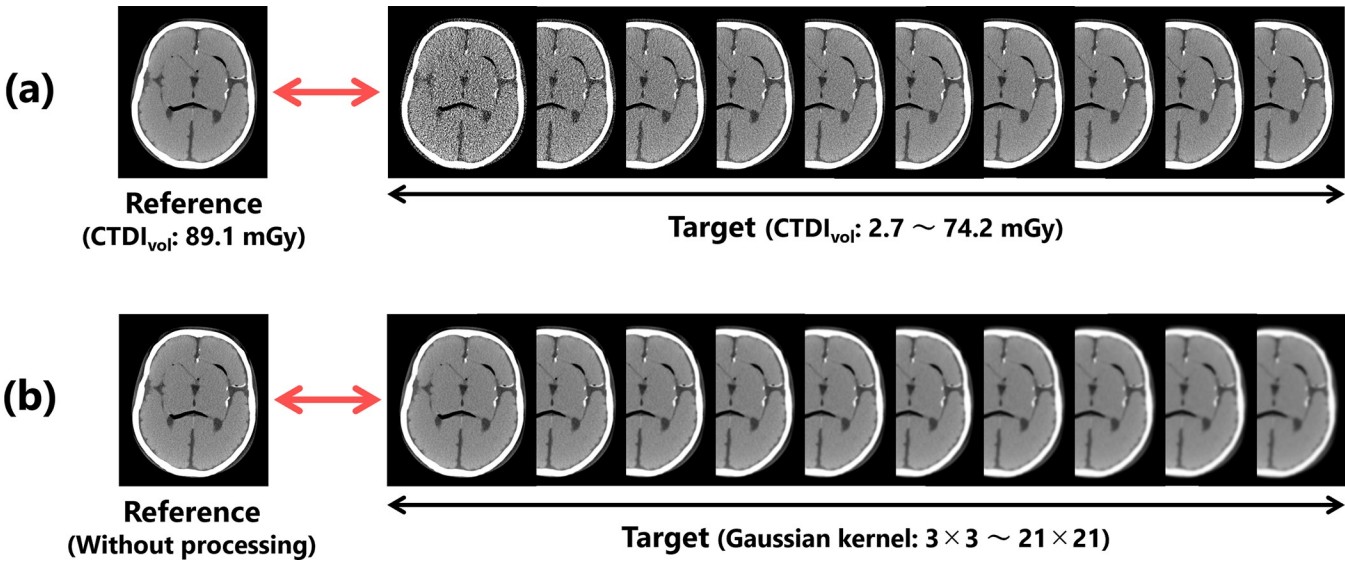

**Fig 2. Image data set used for the validation.** Target images with (a) different noise characteristics and (b) different resolution characteristics.

CTDI$_{vo}$ of the target images ranged from 2.7–74.2 mGy (10 different mGy values: 2.7, 5.5, 13.7, 21.9, 27.4, 35.6, 41.0, 53.4, 59.4, and 74.2).

Other target images were generated by blurring the reference image with a Gaussian filter to change its sharpness (Fig 2(B)). The filter kernels for the processing were set to 10 different sizes, ranging from 3×3 to 21×21 (3, 5, 7, 9, 11, 13, 15, 17, 19, and 21). The parameter $\sigma$ of the Gaussian filtering was adjusted based on the kernel size $k$ using the following equation [40]:

$$\sigma = 0.3 \times \left(\frac{k}{2} - 1\right) + 0.8. \tag{3}$$

To demonstrate the usefulness of the proposed index, we prepared a set of target images by performing several affine transformations on the reference image that was used for validity verification. All the images used in the verification of the usefulness were obtained under the same dose level without Gaussian processing, implying that the images that had been used in evaluation did not inherently differ in terms of image quality. The affine transformations that were used were translation (11 types: 2, 4, 8, 12, 16, 20, 24, 28, 32, 36, and 40 pixels), rotation (11 types: 1, 3, 5, 10, 15, 20, 25, 30, 35, 40, and 45°), scale conversion by reduction (5 types: 0.75, 0.8, 0.85, 0.9, and 0.95 times), and enlargement (5 types: 1.05, 1.1, 1.15, 1.2, and 1.25 times), as shown in Fig 3.

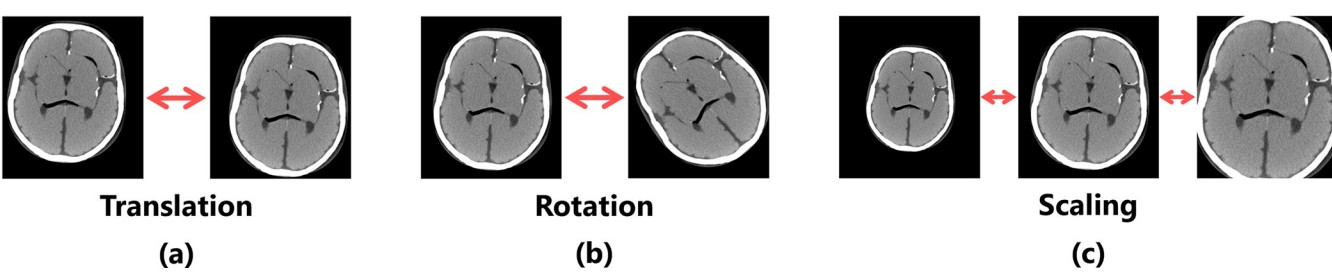

Translation
(a)

Rotation
(b)

Scaling
(c)

**Fig 3. Overview of the image set used for usefulness verification.** Targets are images with (a) different levels of translation, (b) different degrees of rotation angles, and (c) different levels of scales.

## Verification of proposed index

To verify the validity of the proposed index, its behavior was examined compared with that of the SSIM, which was used as an external criterion [42, 43]. If the proposed index correlated with the SSIM with respect to changes in image quality, the proposed index could be regarded as an objective quantitative assessment value related to subjective perceptions of image quality. The proposed index was applied to the evaluation of target images with affine transformations, which demonstrated its usefulness and confirmed its correlation with the SSIM. The effects of processing steps that did not affect the image quality were analyzed to assess the robustness of the proposed quantitative values against translation, rotation, and scale conversion, which negatively affect the accuracy of full-reference type metrics. Under these conditions, if there was no correlation between the changes in the proposed index and SSIM, the proposed index would be considered more robust than the SSIM and regarded as a more useful image quality indicator. In this validation, we conducted a significance test using Pearson's product-moment correlation coefficient with a significance level of 5% for statistical analysis of the correlation with external standards.

A self-made program using the programming language Python 3.9.13 and image processing library OpenCV 4.6.0 was used to compute the proposed index and SSIM. Various parameters can be arbitrarily set for the feature detection algorithm; however, in this verification, we adopted the default values provided by the OpenCV library [44].

## Results

### Correspondence between keypoints

Fig 4 shows the loss in the correspondence between keypoints with changes in the exposure dose. Changes in image quality can provide a qualitative understanding with respect to feature values not matching appropriately, even at the same local point in an image. In addition, the number of detected keypoints differed under each processing algorithm. Table 1 lists the number of keypoints that were detected and the number that matched with the corresponding points for each algorithm.

### Evaluation of validity of proposed index

Fig 5(A) shows the results of evaluating the validity of the proposed index on several target images at different doses. The horizontal axis represents the SSIM, which is the external reference, and its value changes depending on the exposure dose ($CTDI_{vol}$) at which the images are captured. A higher imaging dose results in a better, SSIM of the target image. The vertical axis on the left represents the quantitative value of the proposed index and that on the right represents the value of the PSNR, which was included for comparison purposes. The results calculated by three algorithms were plotted. The correlations were examined using the Pearson's correlation coefficient. A significant correlation ($p < 0.05$) was observed between the behavior of the SSIM and that of the proposed index with respect to changes in the imaging dose for all algorithms. Table 2 summarizes the correlation coefficients of the evaluation values with respect to the SSIM.

Furthermore, Fig 5(B) shows the results of evaluating the images with the proposed index, where the sharpness of the images was modified using the Gaussian filter. The value of the SSIM on the horizontal axis changes depending on the processing strength of the Gaussian blurring. Similar to the behavior in response to changes in the amount of image noise shown in Fig 5(A), there was a significant correlation ($p < 0.05$) between the changes in the quantitative values of the SSIM and those of the proposed index for all algorithms.

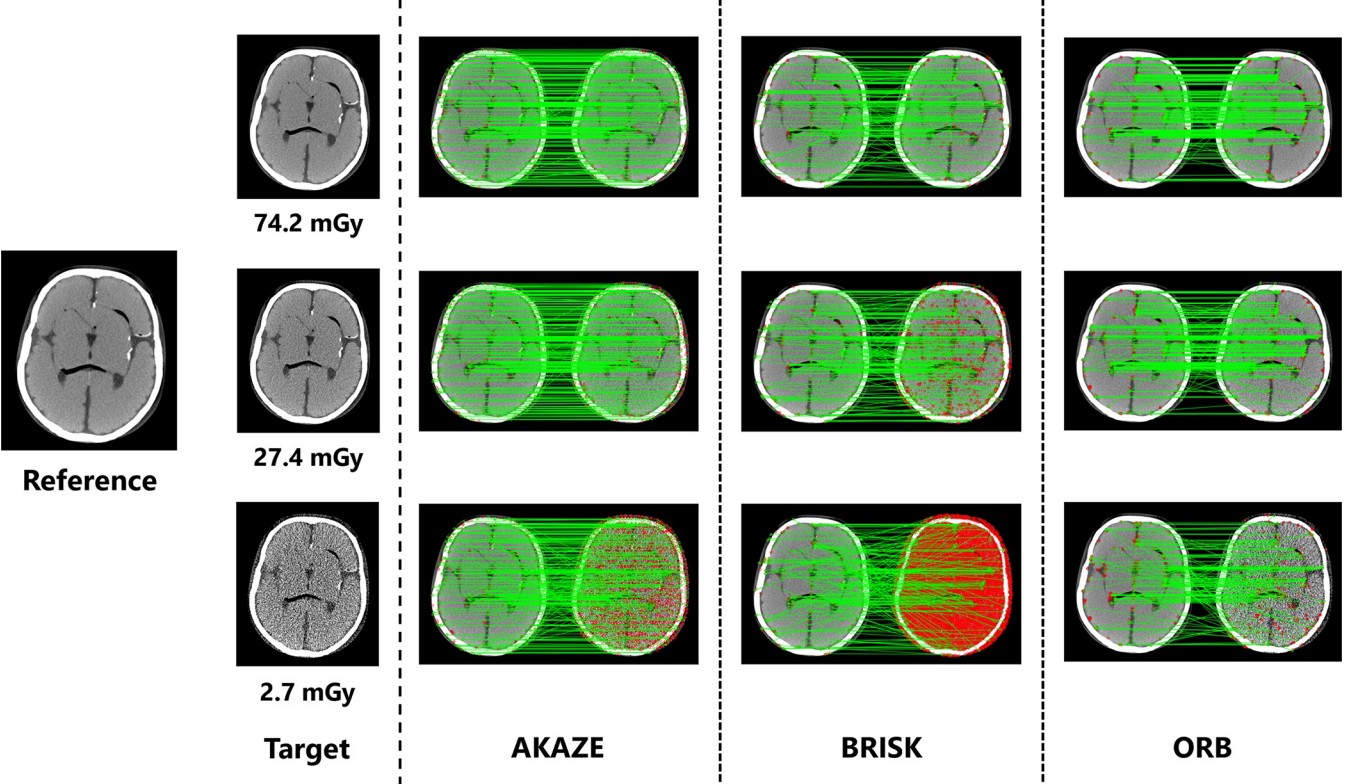

**Fig 4. Changes in feature point correspondences with variations in image noise levels.** When the image quality changes, the features no longer match properly, even at the same position in the image. The left and right sides represent the reference and target images, respectively, for each algorithm. The red dots represent the keypoints extracted for each image, and the green lines represent matches between the keypoints.

The measurement results for the computational complexity of the proposed index are listed in Table 3. All measurements were performed independently in the same hardware environment to avoid the influence of computer usage, and their means and standard deviations were calculated. The results demonstrate the superiority of the ORB algorithm, indicating its ability to perform quantification of evaluation values and visualization of keypoints quickly. Although the BRISK algorithm required a relatively longer time of approximately 300 ms, we consider

**Table 1. Summary of the number of extracted keypoints and number of matching points under different algorithms for images with varying dose levels.**

| CTDI$_{vol}$ [mGy] | AKAZE | | BRISK | | ORB | |
|---|---|---|---|---|---|---|
| | Keypoints | Matchings | Keypoints | Matchings | Keypoints | Matchings |
| 2.7 | 1762 | 259 | 8475 | 177 | 500 | 177 |
| 5.5 | 1092 | 272 | 6122 | 201 | 500 | 183 |
| 13.7 | 598 | 288 | 2719 | 194 | 500 | 235 |
| 21.9 | 523 | 291 | 1360 | 181 | 500 | 253 |
| 27.4 | 499 | 299 | 963 | 182 | 500 | 260 |
| 35.6 | 445 | 297 | 602 | 179 | 496 | 274 |
| 41.0 | 461 | 297 | 517 | 171 | 497 | 278 |
| 53.4 | 411 | 299 | 378 | 166 | 497 | 287 |
| 59.4 | 402 | 294 | 344 | 175 | 500 | 274 |
| 74.2 | 395 | 294 | 305 | 172 | 500 | 297 |

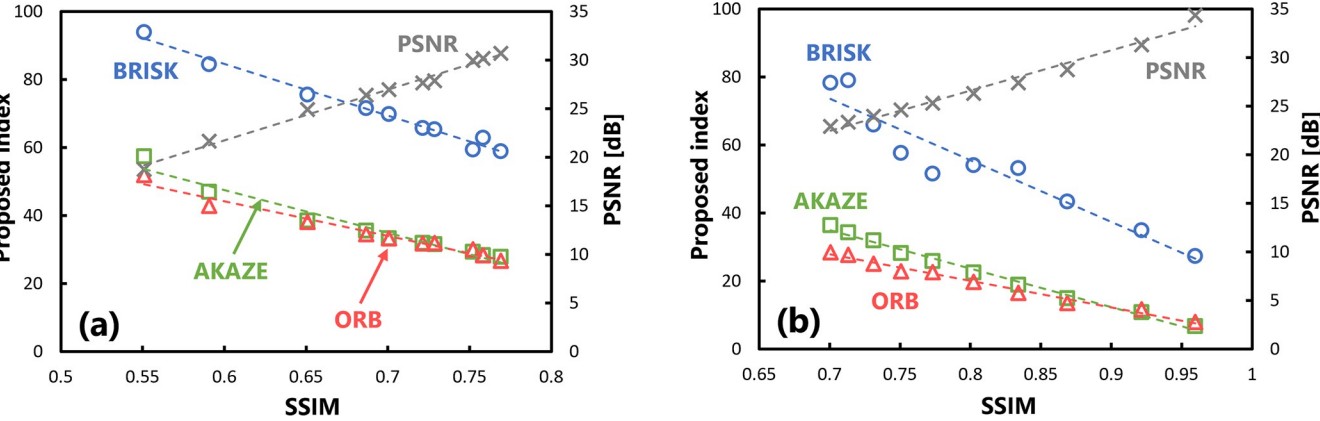

**Fig 5.** Scatter plots of the SSIM and proposed index obtained in the validation when there were changes in the (a) exposure dose levels and (b) intensity of Gaussian processing. Statistically significant correlations were obtained for all algorithms in both conditions.

this value insignificant in terms of the practical computation time. The difference among the feature analysis algorithms depends on the number of extracted keypoints, as shown in Table 1. Reducing this number through hyperparameter tuning will allow for even faster execution.

### Assessment of usefulness of proposed index

Next, the usefulness of the proposed index in evaluating target images subjected to affine transformations was evaluated. The results of calculating the SSIM and proposed index by translating the target images are plotted in Fig 6(A). Each plot represents the values obtained when the image is translated into any pixel. The value of the SSIM on the horizontal axis varies depending on the degree of translation; a larger shift results in a lower SSIM. Regarding the variations in the SSIM and proposed index with changes in translation, there was no significant correlation between the indices under the BRISK and ORB algorithms. Although there was a significant correlation between them under the AKAZE algorithm ($p < 0.05$), the results were stable, with only small changes. Table 4 summarizes the correlation coefficients of the evaluation values with respect to the SSIM.

Fig 6(B) shows the results of calculating the SSIM and proposed index after rotating the target image by affine transformation. There was a significant correlation ($p < 0.05$) between the change in the SSIM and proposed index with respect to the variation in the amount of rotation. However, although they correlated under the ORB algorithm, the change in the value with respect to the SSIM was smaller than that under the other algorithms.

**Table 2. Summary of the correlation coefficient under different algorithms for images with varying image quality.**

| Algorithm | Correlation coefficient | |
|---|---|---|
| | **Changing the exposure dose (Fig 5(A))** | **Changing the unsharpness intensity (Fig 5(B))** |
| AKAZE | -0.981 ($1.690 \times 10^{-7}$) | -0.995 ($4.721 \times 10^{-10}$) |
| BRISK | -0.992 ($3.092 \times 10^{-9}$) | -0.954 ($7.509 \times 10^{-6}$) |
| ORB | -0.983 ($9.080 \times 10^{-8}$) | -0.993 ($1.589 \times 10^{-9}$) |
| (PSNR) | 0.995 ($3.053 \times 10^{-10}$) | 0.989 ($1.526 \times 10^{-8}$) |

The numbers in parentheses are p-values obtained by Pearson's correlation analysis. The significance level is set to $p < 0.05$.

**Table 3. Comparison of calculation times when using each algorithm.**

| Algorithm | Calculation time [ms] | | |
|---|---|---|---|
| | Output of quantitative value | Visualization of feature points | Total |
| AKAZE | 62.2 ± 1.4 | 41.7 ± 4.1 | 103.9 ± 4.3 |
| BRISK | 233 ± 2.1 | 78.9 ± 1.3 | 311.9 ± 2.5 |
| ORB | 17.9 ± 0.0 | 29.5 ± 3.3 | 47.4 ± 3.3 |
| SSIM | 98.9 ± 11.0 | – | 98.9 ± 11.0 |

Fig 6(C) and 6(D) show the evaluation results for the scaled target image. Regarding changes in the SSIM and proposed index with respect to the reduction rate, no significant correlations were observed between them under the AKAZE and ORB algorithms (Fig 6(C)). Furthermore, there was no significant correlation between the proposed index under the ORB algorithm and SSIM with respect to the change in enlargement rate (Fig 6(D)).

## Discussion

We have introduced a novel quantitative metric in the context of medical image analysis under the hypothesis that the distances between the feature descriptors of the corresponding keypoints could be utilized as an indicator for image quality evaluation. The quantitative values

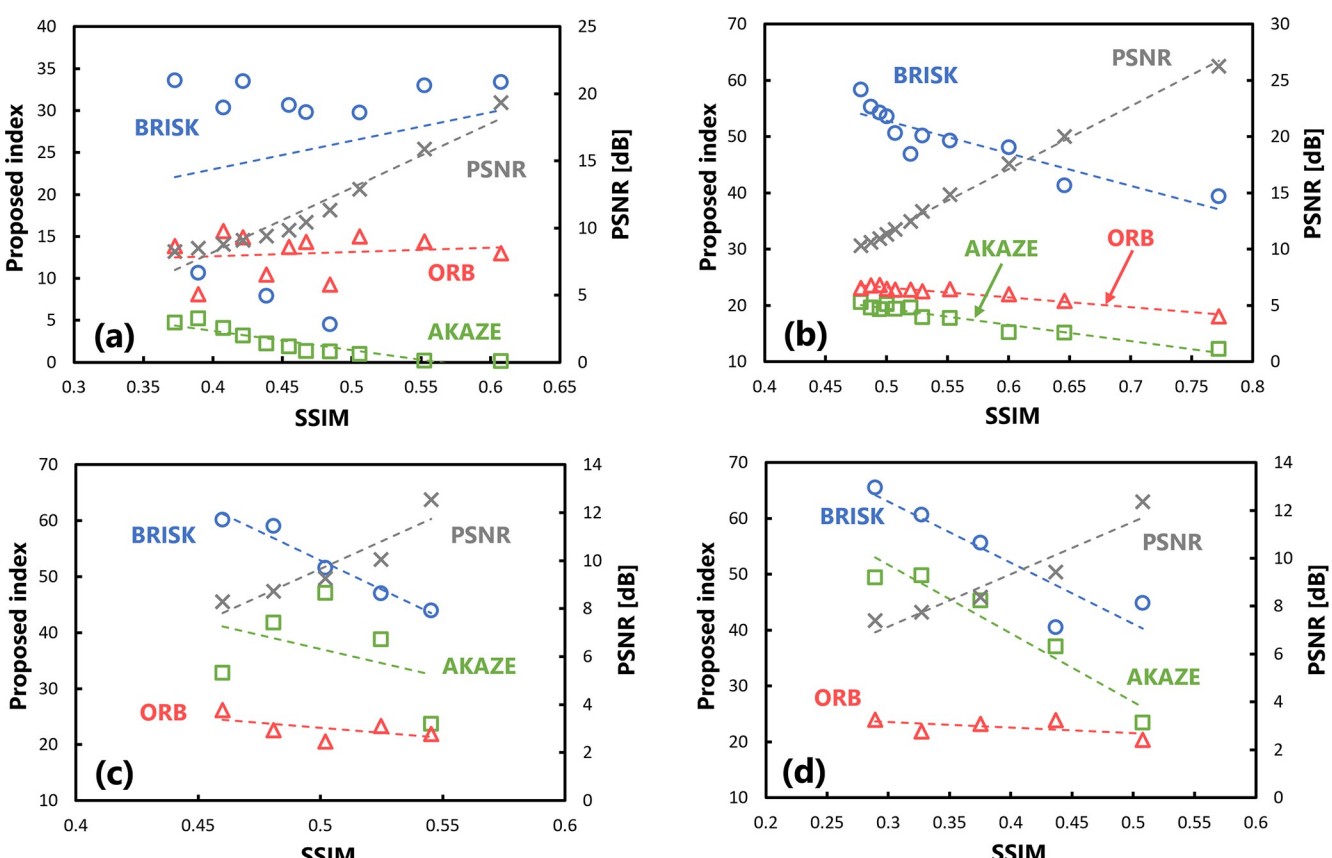

**Fig 6. Scatter plots of the SSIM and proposed index in the usefulness verification.** (a) There are changes in the amount of translation, (b) there are changes in the angle of rotation processing, (c) several downscaling transformations have been performed, and (d) several enlargement processing steps have been performed.

**Table 4. Summary of the correlation coefficient under different algorithms for images with varying image quality.**

| Algorithm | Correlation coefficient | | | |
|---|---|---|---|---|
| | Changing the amount of translation (Fig 6(A)) | Changing the angle of rotation (Fig 6(B)) | Changing the reduction rate (Fig 6(C)) | Changing the enlargement rate (Fig 6(d)) |
| AKAZE | -0.915 ($4.429 \times 10^{-5}$) | -0.968 ($3.774 \times 10^{-7}$) | -0.378 ($2.700 \times 10^{-1}$) * | -0.963 ($1.920 \times 10^{-3}$) |
| BRISK | 0.211 ($3.093 \times 10^{-1}$) * | -0.892 ($1.358 \times 10^{-4}$) | -0.982 ($4.630 \times 10^{-4}$) | -0.904 ($1.220 \times 10^{-2}$) |
| ORB | 0.140 ($3.520 \times 10^{-1}$) * | -0.973 ($1.622 \times 10^{-7}$) | -0.583 ($1.600 \times 10^{-1}$) * | -0.566 ($1.700 \times 10^{-1}$) * |
| (PSNR) | 0.969 ($3.289 \times 10^{-7}$) | 0.998 ($4.484 \times 10^{-13}$) | 0.923 ($8.005 \times 10^{-3}$) | 0.956 ($2.694 \times 10^{-3}$) |

The numbers in parentheses are p-values obtained by Pearson's correlation analysis. The significance level is set to $p < 0.05$. Asterisks indicate items for which no significant correlation was observed.

obtained using the proposed index accurately represent that a smaller value means, the target image is more similar to the reference image, effectively indicating the superior quality of these images.

In our evaluation, the variations in the noise and resolution characteristics of the images were investigated, and the proposed index consistently provided objective quantitative values for these physical image quality changes. The significant correlation observed between the index and SSIM demonstrated the applicability of expressing perceptual quality in terms of objective quantitative values using the proposed index. Such changes in physical image quality may arise in actual clinical tasks when adjusting imaging conditions or modifying reconstruction processing. Therefore, the index is considered to have validity as a practical quality evaluation indicator for medical images, and is also expected to provide stable results when analyzing different variations in image quality. In addition, we found that changes in image quality led to changes in the feature descriptors of the corresponding keypoints. This finding suggests the potential utility of a local keypoint feature in image analysis and quality assessment.

The proposed index was also evaluated using target images subjected to several affine transformation processes that did not inherently alter the fundamental image quality. In the results, the proposed index tended to exhibit a lack of correlation with the SSIM, suggesting the robustness of the proposed index against variations such as translation, rotation, and scaling. The analysis results demonstrate the merits of the proposed index, which can be clearly differentiated from the SSIM, where the values decrease even though the image quality has not changed. This not only highlights the ability of the proposed index to perform structural similarity analysis similar to SSIM, but also demonstrates its usefulness in addressing the concerning aspects of existing reference-based metrics. In our examination, an analysis was performed using three different feature detection algorithms; the ORB algorithm exhibited the best performance. The quantitative values obtained using this algorithm remained stable for all affine transformation processes, indicating that the reliability of the output was ensured. This was partly attributed to the limited number of extracted keypoints, implying that carefully selecting keypoints with high feature intensities may have rendered them less susceptible to the subtle changes in feature values induced by the affine transformations.

One advantage of image analysis using the proposed index based on an image feature is that the corresponding keypoints across images can be visually represented. We believe that this perspective creates a clear distinction from conventional image quality evaluation techniques that rely solely on quantitative values. This enables a straightforward comprehension of structures and local regions with features that may be lost owing to image quality degradation. Our approach comprehensively assesses quality changes in local regions and provides the results as a holistic indicator combined with quantitative values, which is expected to enable more

overall image quality evaluation. Visualizing the feature changes between images using the proposed index provides clarity regarding the specific keypoints that are affected by degradation factors. Based on this information, it would be possible to find new approaches and applications in image analysis that can perform tasks such as the optimization of imaging conditions in medical imaging and development of processing to mitigate quality deterioration.

In actual situations, the required quality of diagnostic images depends on specific clinical questions being addressed. This validation focused on optimizing CT imaging with the head phantom, and how to address specific clinical questions is a future challenge. However, this index can describe how changes in image quality specific to certain imaging conditions lead to a loss of particular features in particular areas; for tasks such as the detection of microcalcifications in mammography or small hemorrhages in neuroimaging examinations, it may be possible to consider whether enhancement processing can be performed without altering the local features. In addition, in diagnostic task scenarios with low-contrast signals such as mass shadows, it may be useful for exploring acceptable levels of dose reduction and optimal processing intensities for denoising. Further investigations are required to determine whether this approach can be seamlessly applied to various clinical tasks, taking into account the relevance to observer reading processes and decision-making.

As the proposed methodology does not have normalized theoretical values such as the SSIM, it might be difficult to use quantitative values for quality assessment when different feature vectors are obtained depending on the variations in imaging systems or image quality conditions. For example. in CT imaging, variations in the physical characteristics of images occur with adjustments in parameters such as tube voltage modulation or slice thickness settings, resulting in changes in the feature descriptors of keypoints. As a result, the output values of the index also change, so the results obtained should be carefully interpreted. To generalize the applicability and effectiveness of the proposed index further, it is essential to analyze its properties in relation to images with varying contrasts and structures, thereby necessitating a broader evaluation in another medical modality. Additionally, validating the metric using different slices for the reference and target images can help to explore its potential as a sub-reference metric for evaluations. We believe that this expectation would help to reduce the influence of patient-to-patient anatomical differences on this metric when considering its use in patients. It is clear that this perspective is unexplored, which highlights the need for preparation for clinical applications.

Considering the accessibility of this assessment framework, we exclusively employed feature detection algorithms that are implementable with the OpenCV library; a more accurate index can potentially be obtained by combining multiple algorithms that are unrestricted by this constraint. In the future, we plan to continue validating and exploring the potential of the proposed index to be more widely applicable by experimentally verifying its use under various algorithm combinations and parameter optimizations. Furthermore, the idea of validating the robustness in more detail through comparative verification based on reference-based metrics, such as the Complex Wavelet-SSIM, which efficiently detects unstructured distortions, is intriguing and crucial, and will build upon findings of the present study [45, 46].

## Conclusions

We have proposed a quantitative metric based on the novel concept of utilizing the distance between the feature values of the corresponding keypoints as an image quality evaluation index. The validity and usefulness of the proposed index were examined using the SSIM as an external reference. The results obtained from evaluating target images with altered noise and

resolution characteristics demonstrated the feasibility of applying the assessment framework to analyze changes in image quality. Moreover, compared to conventional full-reference metrics, such as the PSNR and the SSIM, the proposed index exhibited excellent robustness against translation, rotation, and scale transformations. These findings aid in overcoming the limitations inherent in conventional evaluation methodologies and contribute to image analysis in multicenter settings. Furthermore, the insights obtained from effectively utilizing image feature values for image quality assessment highlighted the potential of the proposed framework in medical image quality evaluation and image analysis. As this new framework does not use machine learning algorithms, it is free from dependencies on hardware or training data. We expect this to serve as a major advantage that will improve the interpretability and explainability of the analysis results, leading to greater versatility. In future research, we will attempt to increase the applicability of the index and incorporate further useful improvements.

## Acknowledgments

We would like to thank Editage (Cactus Communications Co. Ltd., Tokyo, Japan) for English language editing.

## Author Contributions

**Conceptualization:** Sho Maruyama.

**Data curation:** Sho Maruyama.

**Formal analysis:** Sho Maruyama.

**Investigation:** Sho Maruyama.

**Methodology:** Sho Maruyama.

**Project administration:** Sho Maruyama.

**Resources:** Sho Maruyama.

**Software:** Sho Maruyama.

**Supervision:** Sho Maruyama, Haruyuki Watanabe, Masayuki Shimosegawa.

**Validation:** Sho Maruyama, Haruyuki Watanabe, Masayuki Shimosegawa.

**Visualization:** Sho Maruyama.

**Writing – original draft:** Sho Maruyama.

**Writing – review & editing:** Sho Maruyama, Haruyuki Watanabe, Masayuki Shimosegawa.

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
