## [Decision Letter · Decision Letter 0]

30 Jan 2024

PONE-D-23-42003An image quality assessment index based on image features and keypoints for X-ray CT imagesPLOS ONE

Dear Dr. Maruyama,

Thank you for submitting your manuscript to PLOS ONE. After careful consideration, we feel that it has merit but does not fully meet PLOS ONE’s publication criteria as it currently stands. Therefore, we invite you to submit a revised version of the manuscript that addresses the points raised during the review process.

The review report requires substantial revisions to improve clarity and coherence. In the introduction, it is necessary to clearly present the aim/objective and hypothesis before detailing the methods, while eliminating redundant sections such as the outline. The methods section lacks a description of the statistical analysis employed. Moving to the discussion, there is a need to explore the method's applicability in patient settings, considering variations in normal anatomy, and addressing how parameters like kVp-settings and collimation might affect results. Additionally, the discussion should clarify how the method's results correlate with image quality across different clinical questions and highlight its advantages over existing standard methods. Overall, the report would benefit from thorough language editing to rectify numerous typos and grammatical errors, enhancing its readability and comprehension.

We look forward to receiving your revised manuscript.

Kind regards,

Yan Chai Hum

Academic Editor

PLOS ONE

Additional Editor Comments:

Please revise according to reviewer's comments.

Reviewers' comments:

Reviewer's Responses to Questions

**Comments to the Author**

1. Is the manuscript technically sound, and do the data support the conclusions?

Reviewer #1: Yes

2. Has the statistical analysis been performed appropriately and rigorously? 

Reviewer #1: Yes

3. Have the authors made all data underlying the findings in their manuscript fully available?

Reviewer #1: Yes

4. Is the manuscript presented in an intelligible fashion and written in standard English?

Reviewer #1: No

5. Review Comments to the Author

Reviewer #1: Introduction:

- There is a need to present a clear aim/objective and the hypothesis before stating the methods.

- Lines 52-63, is this method description?

- Lines 64-66 are hard to follow. Is this a conclusion?

- The outline in lines 66-69 is not needed

Methods:

- Statistical analysis description is missing.

Discussion:

- There is a need to discuss the use in patients and how differences in normal anatomy might affect the use of this method.

- Could different kVp-settings, collimation, slice thickness, pitch, etc., have changed the results?

- Diagnostic image quality is dependent on the clinical question. How are the results of these tests an index for image quality in different clinical questions? Are there some clinical questions where it is more useful than others?

- What does this method add to image quality assessment compared to today's standard methods?

General:

- The text is hard to follow, and there are some typos and grammatical errors. Thus, there is a need for language editing.

6. PLOS authors have the option to publish the peer review history of their article (what does this mean?). If published, this will include your full peer review and any attached files.

Reviewer #1: **Yes: **Elin Kjelle

---

## [Author Response · Author response to Decision Letter 0]

5 Mar 2024

Reviewer #1

Introduction:

- There is a need to present a clear aim/objective and the hypothesis before stating the methods.

Thank you for your feedback and the suggestion. 

We have extensively revised the final paragraph of the Introduction section, providing a comprehensive overview of the proposed methodology’s objectives and novelty. These modifications aim to enhance the clarity and accessibility of the research introduction.

We have reflected this comment by lines 46–57.

- Lines 52-63, is this method description?

We appreciate your observation and pointing out. 

The description in the “Objective” sentence indeed resembled methodological details, causing confusion in the structure. We have removed the mentioned portion and restructured it so that specific methodology details are now explained solely in the subsequent Methods section.

- Lines 64-66 are hard to follow. Is this a conclusion?

Thank you for your specific feedback. We have removed unnecessary paragraphs and added new sentences to clarify the main contributions of this study.

We have reflected this comment by lines 55–57.

- The outline in lines 66-69 is not needed

Thank you for your advice. 

We have removed the paragraph outlining the structure of the paper. Additionally, we have streamlined the Introduction by eliminating redundant explanations and definitions. These revisions were made with a focus on enhancing readability.

We have reflected this throughout the Introduction section.

Methods:

- Statistical analysis description is missing.

We appreciate the feedback.

We have included a description sentence of the statistical analysis in the Methods section. Please review to ensure that it provides the necessary information for the satisfaction of the reviewers.

We have reflected this sentence by lines 147–149.

Discussion:

- There is a need to discuss the use in patients and how differences in normal anatomy might affect the use of this method.

Thank you for your insightful and valuable comments, which broadens our perspective. 

We have added our argument into the existing text. While acknowledging the need for some validations of how differences in normal anatomy might affect its use, we have considered that it would be difficult to further discuss the regarding additional influence or applications based on the insights obtained in this study. Therefore, we have opted to address the importance of preparing for clinical application and highlight it as a topic for future validation. We appreciate your review of these revisions.

We have incorporated reviewer’s comments by lines 298–302.

- Could different kVp-settings, collimation, slice thickness, pitch, etc., have changed the results?

Thank you for providing specific question. Although this aspect was previously commented in the discussion section, we have now provided more specific explanation. 

If image quality is changed by adjusting the parameters of imaging condition, the results obtained by the proposed method will change. This can be explained based on our results, which examine the effects of dose and blurring in our validations; features of any keypoints may change as the image quality changes. However, to ensure a clearer understanding of the points raised, we have added more detailed explanations to clarify this aspect further.

We have incorporated your comments by lines 288–294.

 

- Diagnostic image quality is dependent on the clinical question. How are the results of these tests an index for image quality in different clinical questions? Are there some clinical questions where it is more useful than others?

Thank you for your insightful perspective and important feedback. We agree with the reviewer's comments. 

Based on the suggestion provided by the reviewer, we have added a new paragraph to discuss this aspect. Considering the relevance to the observer's reading process and decision-making, further investigation is necessary to determine whether the proposed approach can seamlessly apply to various clinical questions or tasks. We have provided examples and outlined future prospects to address this concern.

We have incorporated your comments by lines 275–285.

- What does this method add to image quality assessment compared to today's standard methods?

Thank you for your feedback. We have perceived this as a crucial comment to emphasize the significance of our approach.

One of the clear distinctions between our method and conventional techniques is the ability to visualize keypoints (or local area) where changes in image quality occur; we believe this aspect encourages a better understanding in medical image analysis. To facilitate this understanding, we have added several sentences and modified one paragraph in the discussion section.

We have incorporated your comments by lines 265–270.

General:

- The text is hard to follow, and there are some typos and grammatical errors. Thus, there is a need for language editing.

Thank you for your helpful comments. The submitted manuscript had corrected by a professional editing service; however, we sincerely apologized for any insufficiencies. After revising the points raised by the reviewers, we have resubmitted the manuscript for further proofreading by the service. (We had also attached the reviewer's comments regarding the language in the manuscript.) 

Please review to ensure that the revised manuscript meets the standards of clarity and readability required for publication.

We have reflected this comment the manuscript (minor changes have been made the throughout text).

---

## [Decision Letter · Decision Letter 1]

19 Apr 2024

PONE-D-23-42003R1Image quality assessment index based on image features and keypoints for X-ray CT imagesPLOS ONE

Dear Dr. Maruyama,

Thank you for submitting your manuscript to PLOS ONE. After careful consideration, we feel that it has merit but does not fully meet PLOS ONE’s publication criteria as it currently stands. Therefore, we invite you to submit a revised version of the manuscript that addresses the points raised during the review process.

**Please revise according to reviewer's comments. **

We look forward to receiving your revised manuscript.

Kind regards,

Yan Chai Hum

Academic Editor

PLOS ONE

Additional Editor Comments:

Please revise according to reviewers' comment.

Reviewers' comments:

Reviewer's Responses to Questions

**Comments to the Author**

1. If the authors have adequately addressed your comments raised in a previous round of review and you feel that this manuscript is now acceptable for publication, you may indicate that here to bypass the “Comments to the Author” section, enter your conflict of interest statement in the “Confidential to Editor” section, and submit your "Accept" recommendation.

Reviewer #1: All comments have been addressed

Reviewer #2: (No Response)

Reviewer #3: (No Response)

2. Is the manuscript technically sound, and do the data support the conclusions?

Reviewer #1: Yes

Reviewer #2: Partly

Reviewer #3: Partly

3. Has the statistical analysis been performed appropriately and rigorously? 

Reviewer #1: Yes

Reviewer #2: No

Reviewer #3: N/A

4. Have the authors made all data underlying the findings in their manuscript fully available?

Reviewer #1: Yes

Reviewer #2: Yes

Reviewer #3: No

5. Is the manuscript presented in an intelligible fashion and written in standard English?

Reviewer #1: Yes

Reviewer #2: Yes

Reviewer #3: No

6. Review Comments to the Author

**Reviewer #1**: (No Response)

**Reviewer #2:** Explain how the present manuscript differs from and improves upon these established previous works. Include a study with your survey papers and be sure to highlight any knowledge gaps that have come up recently.

Discuss some major sections in the article like the real time/ practical application of the paper, future perspectives, major contribution in the paper, significance of this study.

In the introduction section, include some other research applications on medical imaging as literature. Some suggestions are: Multi-modal medical image fusion framework using co-occurrence filter and local extrema in NSST domain; An end-to-end content-aware generative adversarial network based method for multimodal medical image fusion; TSJNet: A Multi-modality Target and Semantic Awareness Joint-driven Image Fusion Network; Clustering based Multi-modality Medical Image Fusion; Directive clustering contrast-based multi-modality medical image fusion for smart healthcare system

The authors should test their model with their dataset (not only from the public dataset). The complexity comparison should be made. Cross-Validation is lacking in this paper.

For the validation and best performance of result, use graphical analysis like histogram or intensity profile analysis.

Rewrite conclusion. Add concluding, contribution points in it briefly. Abstract needs to rewritten with better summary of paper.

**Reviewer #3:** This paper presents a simple quality assessment index for CT images. The Hamming distance between the features extracted by the feature extraction algorithm and the corresponding feature points of the reference image is used as the quality assessment result for the image. However, the manuscript is rather confusing in its structural arrangement, the logical expression is not very clear, and it also has the following problems that need to be revised:

The introduction does not make clear what the application context requires of the method. The reason why evaluation indicators such as SSIM cannot be used and new indicators need to be proposed is not well reflected in the introduction.

The design of the experimental section is problematic. The comparison of algorithms in the experimental section does not reflect well the superiority of the proposed quality evaluation method. At the same time, the algorithm lacks data on time complexity.

Insufficient arguments for the conclusions drawn from the experimental results. The conclusion in line 254 that the scores obtained by the algorithmic metrics proposed after the images have undergone affine transformations are not sufficiently correlated with SSIM to indicate that the algorithms are robust is not very convincing. Consideration could be given to validating this point based on the correlation between the results of the algorithm after the affine transformation and the results of algorithms such as CW-SSIM and the like (correlation algorithms that can still efficiently detect unstructured distortions).

There is a lack of explanation about the reason for choosing the feature extraction algorithm. Whether the specific characteristics of CT images were taken into account or whether there were other considerations.

7. PLOS authors have the option to publish the peer review history of their article (what does this mean?). If published, this will include your full peer review and any attached files.

Reviewer #1: No

Reviewer #2: No

Reviewer #3: No

---

## [Author Response · Author response to Decision Letter 1]

14 May 2024

The detailed responses to the reviewers are provided in the attached response letter.

Please refer to the attached file.

Thank you very much for your consideration.

---

## [Editor Report · Decision Letter 2]

21 May 2024

Image quality assessment index based on image features and keypoints for X-ray CT images

PONE-D-23-42003R2

Dear Dr. Maruyama,

We’re pleased to inform you that your manuscript has been judged scientifically suitable for publication and will be formally accepted for publication once it meets all outstanding technical requirements.

Kind regards,

Yan Chai Hum

Academic Editor

PLOS ONE
---

## [Editor Report · Acceptance letter]

30 May 2024

PONE-D-23-42003R2 

PLOS ONE

Dear Dr. Maruyama, 

I'm pleased to inform you that your manuscript has been deemed suitable for publication in PLOS ONE. Congratulations! Your manuscript is now being handed over to our production team.

Kind regards, 

on behalf of

Dr. Yan Chai Hum 

Academic Editor

PLOS ONE